# Implementation of Wearable Sensing Technology for Movement: Pushing Forward into the Routine Physical Rehabilitation Care Field

**DOI:** 10.3390/s20205744

**Published:** 2020-10-10

**Authors:** Catherine E. Lang, Jessica Barth, Carey L. Holleran, Jeff D. Konrad, Marghuretta D. Bland

**Affiliations:** 1Program in Physical Therapy, Washington University School of Medicine, St. Louis, MO 63122, USA; jessica.barth@wustl.edu (J.B.); cholleran@wustl.edu (C.L.H.); jdkonrad@wustl.edu (J.D.K.); blandm@wustl.edu (M.D.B.); 2Program in Occupational Therapy, Washington University School of Medicine, St. Louis, MO 63122, USA; 3Department of Neurology, Washington University School of Medicine, St. Louis, MO 63122, USA

**Keywords:** rehabilitation, motor function, wearable sensors, outcomes, measurement, implementation

## Abstract

While the promise of wearable sensor technology to transform physical rehabilitation has been around for a number of years, the reality is that wearable sensor technology for the measurement of human movement has remained largely confined to rehabilitation research labs with limited ventures into clinical practice. The purposes of this paper are to: (1) discuss the major barriers in clinical practice and available wearable sensing technology; (2) propose benchmarks for wearable device systems that would make it feasible to implement them in clinical practice across the world and (3) evaluate a current wearable device system against the benchmarks as an example. If we can overcome the barriers and achieve the benchmarks collectively, the field of rehabilitation will move forward towards better movement interventions that produce improved function not just in the clinic or lab, but out in peoples’ homes and communities.

## 1. Introduction

Wearable sensor technology to measure human movement is rapidly evolving. Motion-sensing wearable devices continue to get smaller, lighter and have more data storage space, with even better products anticipated in the future. These wearable devices are ubiquitous in the general public in the form of a variety of commercially available, consumer-grade products. Here, we use the term ‘device’ to refer to the wearable unit, ‘sensor’ to refer to the sensors within the device, and ‘wearable device system’ to refer to the collective hardware and software package (see Box 1 for other operational definitions used in this paper). This paper intentionally excludes cell phones as a wearable because cell phones are used for a multitude of purposes and are not routinely ‘worn’ in the same location within and across people. Because of the focus on movement, this paper also excludes wearable sensors designed to measure physiological signals such as heart rate, oxygen saturation, and respiratory rate. Wearable devices include one or more specific sensors, with accelerometers being the most common sensor for quantifying human movement. Many devices also include magnometers, inclinometers, gyroscopes, and light sensors.

Box 1Operational Definitions.*Commercially-available:* device systems that can be purchased through companies or organizations.*Laboratory-available:* device systems developed in one or more labs and not available for purchase by the general public.*Consumer-grade:* device systems that are marketed directly to consumers, intended to be used by anyone.*Research-grade:* device systems that are marketed to researchers and healthcare professionals, intended for use to be managed by someone with specialized training.*Activity:* Execution of a task or action, such as walking, dressing, or bathing [1].*Capacity:* Activity that is assessed in a structured setting, usually with a standardized tool, such as the 10 m walk test or the Box and Block test. Alternate terms are function and functional capacity [1].*Performance:* Activity that is assessed in an unstructured, free living setting. Performance can be measured directly via wearable devices, or via self-report with questionnaires [1].

The potential benefit of wearable sensor technology for physical rehabilitation has been discussed in the literature since the early 2000s [2]. The real promise for rehabilitation is that wearable device systems can measure human movement in real world settings, not just in the structured environment of the clinic or laboratory. People seek physical rehabilitation services to improve their movement in daily life, so rehabilitation professionals need a way to quantify that movement in order to best help their patients. In the published literature to date, two overlapping scientific cohorts are responsible for the majority of the progress with wearable sensor technology. The first cohort includes the engineering groups, who are developing and testing sensors, software, and algorithms. The second cohort includes clinician scientists who are validating sensor-derived metrics in clinical populations and deploying these metrics to answer questions and measure outcomes in rehabilitation studies. While wearable device systems to measure human movement have become more commonplace in research studies, they have not yet been widely implemented in routine clinical practice [3]. Multiple barriers arising both from clinical rehabilitation practice and from the current state of commercially-available devices need to be overcome before there is widespread adoption into routine care. We note that these barriers and their relative importance may vary somewhat around the globe.

The purposes of this paper are to: (1) discuss the major barriers in clinical practice and available wearable sensing technology; (2) propose benchmarks for wearable device systems that would make it feasible to implement them into clinical practice across the world; and (3) evaluate a current wearable device system against the benchmarks as an example. This field is in its early stages. The evolution of the telephone is a good analogy for our field. Early telephones look and function nothing like the sleek, smart phones we use today, with advancements in telephone technology spread out over more than 100 years. This paper is intended as a small step, specifically focused on helping the collective advancement of wearable sensor technology from research groups into routine clinical practice. Our larger goal is to move the field of physical rehabilitation forward towards better interventions that produce improved movement performance in homes and communities, not just improved capacity in our clinics and labs.

## 2. The Current Situation in Clinical Care

There are multiple factors influencing physical rehabilitation that may impact the implementation of wearable device systems into routine clinical rehabilitation practice. Many influencing factors are generic, i.e. they influence any type of change in clinical practice. This paper intentionally focuses on two key factors most strongly influencing clinical uptake of wearable sensor technology. These key factors are the time constraints present in everyday clinical practice and the salience of information from wearable device systems by physical rehabilitation professionals. Note that medical- or research-grade devices that are deployed into clinical care must meet all the clinical standards, defined by the International Organization of Standards (ISO/TC 173 for Assistive Products and ISO 10667-1:2011 for Assessment Service Delivery), and evaluated by regulatory agencies. Regulatory approval is a long, challenging process, varying somewhat by geographic location; a discussion of this process is outside the scope of this review.

### 2.1. Busy Clinical Practice Affords Little Time for Anything Else

Across the world, patients are seen in a variety of rehabilitation settings including acute and long-term hospitals, specialized facilities (skilled nursing, inpatient or acute rehabilitation), outpatient clinics, virtual visits, or at home, each with different requirements for productivity. In the United States for example, productivity standards for physical therapists can be upwards of 85% with reimbursement only for billable units [4]. This means that therapists are working under the expectation that upwards of 6 h 48 min of their 8-h workday are devoted to direct patient care, with patients scheduled back to back and little or no time in between. Meeting these standards can be challenging as direct patient care time is often limited by the patients themselves, both within the hospital or inpatient setting and outside the hospitals in outpatient, ambulatory care settings. Common disruptions to direct patient care time in the inpatient setting are delays or interruptions due to hygiene or dietary concerns, and conflicting medical needs. Similar disruptions in the outpatient setting most frequently arise from transportation delays. Figure 1 provides two examples of typical rehabilitation sessions (Figure 1A: inpatient, physical therapy, Figure 1B: outpatient occupational therapy) to illustrate how planned, scheduled direct patient care time does not often equate to actual care time [5,6].

In addition, regardless of the clinical setting or geographic location, physical rehabilitation professionals (physicians, physical and occupational therapists, etc.) in today’s busy clinical environment have many competing priorities above and beyond direct patient care that limit available time [3,7,8,9]. Rehabilitation clinicians have administrative demands related to documentation, addressing patient and family concerns, and coordination of care with other healthcare team members [10,11]. Competing demands on time are often balanced by the rehabilitation professional multi-tasking, providing group treatments (allowable in some settings or countries, but not allowable in others), and completing administrative tasks while providing treatment during a session. Collectively, these pressures limit time and energy to trial new technology, including wearable device systems.

Figure 2A shows an example of current time costs for using a wearable sensor system within clinical practice, with time estimates based on our laboratory protocol [12]. Asking rehabilitation professionals to charge, program, don/doff, process, and share results (both with the patient and other health professionals through an electronic medical record) of wearable device systems is unrealistic if it cannot be completed quickly during a patient treatment session. Figure 2B shows a clinically-feasible time cost that would foster implementation of wearable device systems into routine clinical rehabilitation practice.

The above examples are drawn from the healthcare system in the United States. While other countries might not currently face such severe time constraints, all counties face at least some. Additional challenges faced across multiple continents include under-sourced physical rehabilitation services (i.e. a limited number of professionals available), which may lead to self-imposed pressures for professionals to treat as many people as quickly as possible. Regardless of the unique situations in each country, it is unlikely that these time constraints and competing priorities are going to change in the foreseeable future. For wearable device systems to be widely implemented into rehabilitation care around the globe, the technology needs to fit in seamlessly, minimizing disruption to busy, clinical settings.

### 2.2. Clinicians Are Still Building towards Understanding the Added Value of Wearable Sensor Data for Clinical Rehabilitation Practice

The World Health Organization’s International Classification of Function, Disability, and Health (WHO-ICF) describes three levels of classification for any health condition (disorder or disease), including body function and structure, activity, and participation [1]. Activity is considered the ability to execute tasks or actions and can be subdivided into the capacity for activity, i.e. what a person can do in a structured environment, and performance of activity in daily life, i.e. what a person actually does (see Box 1). Established in the field of physical rehabilitation is the importance of standardized outcome assessments to evaluate change over the course of clinical care [13]. Recommended activity level measures included in rehabilitation clinical practice guidelines around the world nearly always assess capacity, not performance [14]. But as mentioned in the Introduction, individuals who engage in physical rehabilitation services typically seek improvement in movement performance within their daily lives [15]. Assessing the impact of rehabilitation interventions in the context of an individual’s life could, therefore, serve as a primary indicator of effectiveness of rehabilitation interventions. Examples of using performance-level tracking in physical rehabilitation are the use of a single wearable sensor worn at the ankle to track daily walking (e.g., steps/day), or the use of two wearable sensors worn on the left and right wrists to track upper limb activity (e.g., use ratio, which is the relative duration of activity in one limb vs. the other). Objective, performance-level tracking via wearable devices across clinical episodes of care for decision-making, however, is an emerging, yet not established practice.

While there is promise for the adoption of performance-level tracking within clinical populations, several clinical assumptions stand in the way of wide-spread adoption. First, there is the widespread assumption that capacity measures taken in the clinic reflect performance measures in daily life. Capacity is a snapshot of ability at a singular time point in a structured environment, whereas performance measures capture real-world activity that includes the ecological validity of an individual’s free-living condition. Published data on gait speed, an index of walking capacity, and steps per day, an index of walking performance, illustrate this problem. Gait speed, measured in the clinic, generally accounted for 30–45% of the variance in steps/day, leaving up to 70% of the variance in daily stepping unexplained [16,17]. For example, individuals in a recent study with self-selected gait speeds around 0.8 m/s ranged from about 750 steps/day to over 6000 steps/day (see Figure 2B of [18]). Without a direct measure such as those provided by wearable device systems, a clinician has limited insight about walking or other movement performance in daily life. Up to the present time, physical rehabilitation clinicians have had to rely primarily on self-report measures to quantify the amount, frequency and duration of movement outside of clinical services. Unfortunately, self-report measures have been shown to lack consistency with more direct assessments [19,20]. Thus, wide-spread adoption of wearable device systems into routine clinical rehabilitation practice will provide new, important information for clinical management.

The second clinical assumption is that a change in an individual’s capacity is equivalent to a change in that individual performance in daily life. Over the past few years, multiple reports have now demonstrated discrepant outcomes in capacity and performance over the course of research and clinical interventions [21,22,23,24]. Each of these aforementioned belief barriers will require educational strategies tailored towards rehabilitation professionals to improve adoption of performance tracking with wearable device systems within clinical populations.

The great news is that emerging data exist demonstrating the utility of performance-level tracking at improving a variety of outcomes and monitoring the effects of disease processes on movement longitudinally [25,26]. Performance monitoring as part of research interventions has been effective at increasing daily stepping, improving daily physical activity, and reducing sedentary time in healthy populations (see [27] for review). In clinical populations, performance tracking has been effective at improving functional outcomes including walking endurance [28,29] and daily walking activity [30,31] in the lower limb, and has proven to be sensitive to changes in real-world upper limb use across both research and clinical interventions [22,32]. Further, tracking has also been sensitive to the degradation of movement in daily life in both static [33] and progressive [34] disease processes. Advantages of having performance-level measurements from electronic, internet-connected wearable device systems include: (1) the storage of data in secure, remote databases; (2) the ability to analyze these measurements across large data sets; and the subsequent ability to recommend specific actions based on the results of these measurements. Increased adoption of performance monitoring in the clinical environment could provide the big data required to elucidate the full potential of wearable sensing technology.

## 3. The Current Situation with Wearable Device Systems

### 3.1. Commerically-Available, Consumer-Grade Device Algorithms Have Limited Accuracy in Disabled Patient Populations

A major barrier for widespread clinical adoption of wearable sensor technology is that the most accessible wearable device systems, those marketed directly to consumers, have questionable accuracy in rehabilitation populations. Using terminology set out in the V3 framework [35], the problem is not in the verification of the sensor itself, but rather lies in the analytic and clinical validation of the algorithm. People seeking rehabilitation services often do not move normally, such that the algorithms programmed into consumer-grade devices are inaccurate in identifying or quantifying their movement [36,37,38,39,40,41]. Continuing with our mainstream metric of walking performance, steps/day has been evaluated across many consumer-grade devices. Studies have evaluated the accuracy of these devices across a variety of functional activities and environmental settings, at different placements on the body, and across a wide range of abilities. There was wide variability in the accuracy of these devices in individuals with normal gait speed [42,43]. Furthermore, in individuals who utilize assistive devices (e.g., cane, walker) [44,45,46], walk with slower speeds (e.g., < 0.8 m/s) [36,37,38,39,40,41], or have interruptions in continuous walking [47,48], even higher levels of inaccuracy have been identified. The Fit-Bit Ultra (Fitbit Inc., San Francisco, CA, USA) consumer-level device, for example, has been shown to systematically under-estimate steps for individuals with a diagnosis of stroke and traumatic brain injury over a 2-min walk test, with greater inaccuracy for those who took less steps per minute and those that walked ≤ 0.58 m/s [41]. As many individuals who seek physical rehabilitation walk at slower speeds, this poses a major barrier for accurate, objective monitoring. In contrast to the consumer-grade devices, commercially-available, research-grade sensors such as the Step Activity Monitor (Modus Inc., Edmonds, WA, USA) have demonstrated strong reliability and accuracy across varied levels of abilities, including differing medical diagnosis, variable gait speeds, and use of assistive devices [41,49,50,51,52,53,54] with limited data across differing environmental conditions [40,54,55,56]. Though this research grade device is accurate and reliable in individuals with physical impairments, the device may lack key features that would be essential for widespread clinical adoption (see Section 4 below). Unfortunately, this discussion is an example of only one variable derived from a wearable device. When additional variables are examined, the depth of work to integrate these devices into clinical practice grows exponentially.

### 3.2. Research-Grade Device Systems Are Expensive and Not yet Clinician- and Patient-Friendly

Most research-grade devices work in conjunction with software systems that require a separate computer program to set-up, download and examine the data. To be feasible in the clinic, every rehabilitation professional would need the necessary computer program loaded onto their laptop or other computer they use for clinical care, which would be expensive. In addition, cost of devices for multiple patients and multiple limbs (e.g., wear one device on each wrist to measure upper limb performance) could quickly make wearable sensor technology unreachable for most clinics. Beyond the cost, current output from research-grade wearable device systems is not easily accessible in a timely manner for rehabilitation professionals or for their patients. Current research-grade systems require training to use. Training involves both how to use the device (e.g., turning on/off, charging, specific requirements for sensor placement wear) and how to use the system (e.g., programming the device, uploading/storing data, translating device output into clinically relevant values). For wearable device systems with proprietary software algorithms, only the default outputs (or variables) are available. If the clinician needs variables beyond the defaults, they would need a research colleague to write code and extract them. Research-grade system outputs (default or otherwise, see Table 1 for variables summary) are difficult to provide quickly during or outside of a treatment session. Furthermore, if the rehabilitation clinician has taken the effort to extract the data and variables, the output is not yet patient-friendly. Researchers have tried to bridge this gap by transforming outputs into patient-friendly graphs, but the process is cumbersome and has not yet yielded improved results [57]. For the clinically-important information derived from the sensors to be widely utilized by patients and clinicians, wearable device systems will likely need to be less expensive, continuously streamed [58], and on an accessible consumer-based platform [9]. Wearable device systems will need to be compatible with and seamlessly integrated into the electronic medical record to be readily adopted [59] and to contribute to the quality and effectiveness of rehabilitation services.

### 3.3. Standardization of Output Variables in Research Is Limited to Date, with Much Work to Do

The pathway to routine clinical rehabilitation implementation for wearable sensor technology is not just impeded by current clinical care and available devices, but also by the state of the science about the output variables. While it is clear that measurement of outcomes is essential in both research and clinical practice, the pathway to an established outcome measure or sensor-based variable is a long, hard one [60,61]. It is critical that measures and variables are thoroughly vetted, since scores obtained may be used to make diagnostic and rehabilitation management decisions. Prior to any measure being routinely implemented into clinical rehabilitation care, multiple studies evaluating the psychometric properties and clinical utility of the measure are necessary [60,61,62,63]. Beyond the verification of sensor signals [35], variables must first demonstrate reliability, or consistency of results obtained, indicating that one can trust that the obtained value is stable. Validity is the second hurdle, with multiple layers of validity. These layers, in hierarchical order, include: (1) face validity (does the variable appear to capture the underlying construct); (2) content validity (does the variable adequately capture/sample the underlying construct); (3) criterion-related validity (how does the variable agree with other measures or the gold-standard measure of the same construct); and (4) construct validity (how well does the variable measure the construct). If a sensor-based variable were to be used to make diagnostic decisions, discriminitive validity (how does the variable distinguish between those with and without a specific condition) and predictive validity (how well does the variable predict future outcome) would also need to be demonstrated in the population of interest. Responsiveness is the third and final hurdle and includes the ability of the variable to capture the range of the construct and its sensitivity to change over time.

Table 1 is a sample list of variables proposed in the literature to measure important rehabilitation constructs that might be used for diagnostic or clinical decision-making. These variables have largely been derived from data captured on research-grade devices and calculated with custom software developed in laboratories. The variables are intended to capture different movement constructs that may be of interest to physical rehabilitation clinicians. We have excluded most measures of general physical activity (e.g., caloric expenditure), except steps/day, since general physical activity measures are an extensive topic in their own right. Variables are presented based on their intention to quantify movement at either the lower (generally captured via one sensor on one leg) or upper limb (generally captured with one sensor on each arm). Based on the available data, variables are marked as to their exploratory status and how quality and quantity of data are related to reliability, validity, and responsiveness.

There are three key points to take away from this table. First in looking at the left-hand column, there are many variables with different names and often different formulae that may be capturing similar or related constructs of movement. This is both good and bad. Similarity is good because it signifies that the movement construct is considered important by multiple groups. Different names and different formulae are bad for future progress because they make comparisons across studies, samples, and populations difficult. Second in looking at the next two columns, exploratory data are available for many variables. This highlights the creativity that will be needed to really understand movement constructs, variables, and relationships with clinical practice. Third and most importantly in looking at the right-hand columns, there is a tremendous amount of work needed for most of these variables to demonstrate the reliability, validity, and responsiveness necessary for adoption into routine rehabilitation clinical practice. Only one variable, steps/day has sufficient established psychometric and clinical utility to make it worthy of recommendation to clinicians, i.e. evidence of strong, stable psychometric properties across multiple studies with large samples. Another variable, the use ratio for the upper limb, is making good progress towards this goal, after its initial proposal some 20 years ago [81]. Since many of the other variables are only recently proposed, it may be a long time before needed data are available. We are hopeful that once one or a few variables are deployed into routine care, the process for deploying other variables may be accelerated.

To achieve standardization of variables derived from wearable devices, our field will need to communicate and collaborate on issues such as: (1) sensor placement on the limbs; (2) filtering algorithms for raw data; and (3) sensor-independent reporting of variables (e.g., gravitational units and not activity counts, which vary by manufacturer). Comparisons across studies are often impossible due to these issues [97]. For example, it is unfortunately not possible to compare upper limb activity levels in a small sample of children with Duchene Muscular Dystrophy [98] to a larger, referent sample of typically developing children [83]. The two studies evaluated conceptually-similar variables, but used different calculations, different sensors, and different filtering algorithms. Going forward, it would be of great benefit to develop open-access databases for normative or referent data, in order to compare those with and without specific health conditions. Further progress might be made by pooling or sharing currently available datasets for meta-analyses.

### 3.4. Different Clinical Populations Will Need Different Metrics for Clinical Decision-Making

Much of the variable development summarized above has occurred in the stroke rehabilitation population. While individuals with stroke represent a substantial portion of the world-wide physical rehabilitation population, there are many other clinical populations that could benefit from the ability to capture motor performance in daily life. Given the heterogeneity of physical rehabilitation populations, it is highly likely that different clinical populations will need different wearable-derived variables for clinical decision-making [99]. Important sensor variables developed for one population may not be clinically relevant for another population. For example, the use ratio is an upper limb variable reflecting the relative time one limb is active compared to the other [81,100]. The use ratio has clear clinical relevance in rehabilitation populations with asymmetric effects on the limbs, such as stroke, hemiparetic cerebral palsy, limb amputation/prosthetic use, and recovery from unilateral upper limb surgery. It has little clinical relevance, however, for those with very mild or no asymmetries in motor abilities, such as children with Duchene’s Muscular Dystrophy or hyperactivity disorders, or for adults with some brain injuries, and many spinal cord injuries. The ultimate goal of any clinical assessment, including variables derived from wearable devices is that a value or score on the variable informs clinical-decision making, such that without this value or score, a different clincial decision would have been made and a worse outcome might have occurred. Thus, our challenge going forward extends beyond establishing reliability, validity, and responsiveness for wearable-derived variables towards demonstration of how the score or value can change clinical practice.

### 3.5. Special Considerations for Complexity in Some Populations

#### 3.5.1. Children

Wearable device systems can be a powerful assessment tool for children as objective assessment of their motor activity is difficult. Direct behavioral observation can be used to objectively assess children in their typical environments but that is costly, time-consuming, and only feasible in research. Clinic-based assessments are time consuming, require trained experts, have subjective components, and require the children to be assessed outside of their familiar environment. Furthermore, children under 10 or children with special needs cannot always follow commands or accurately report on their performance [101]. Proxy-reports from parents or teachers have limited accuracy [101,102]. Wearable devices, therefore, can provide pediatric rehabilitation clinicians with real-world, objective performance data on their patients [103,104,105,106]. Wearable devices have been used to research the behavior of children with attention deficit-hyperactivity disorder (ADHD) in the classroom [107]. In autism spectrum disorder research, wearable devices have been used to study sleep patterns, stereotyped behaviors, and metabolic disease risk [108,109,110,111,112,113,114]. Researchers have also used wearables to identify infants with developmental delays and infants with movement disorders associated with cerebral palsy [115,116] and children with ADHD [105]. While their value for pediatric physical rehabilitation is clear, two special considerations for pediatric wearable device system use are the data collection protocol and the physical design of the device.

Particularly in pediatric studies, there has been wide variation in data collection procedures with respect to device placement and wearing duration. Devices are placed on the waist, ankle, one or both wrists, or a combination of these sites. The issue of device placement is further complicated by the different, and often more intense, movement activity in children (e.g., swinging, climbing, and sliding on playgrounds) compared to adults. While multiple sensors can quantify movement in ways a single sensor cannot, multiple sensors create a greater burden for the wearer, particularly on infants and small children. Implementation of wearable sensor technology into pediatric clinical practice will require thoroughly vetted protocols that can maximize sensor information and minimize the number of sensors and duration of wear. Thoroughly vetted protocols will also be an important part of convincing parents to allow and promote device wearing in their children.

Routine clinical implementation of wearable devices and systems for pediatric rehabilitation populations will also depend on the physical design of the device and how it is secured to the individual. Size, weight, aesthetics, and ease of donning/doffing, are all key considerations. An ideal wearable device for pediatric patients would be small and light enough for infant wrists and ankles but robust enough to handle the stresses encountered while being worn by a child or teen for prolonged periods. The device must not impede or alter the movements that are being recorded [117]. The strap should be strong, non-absorbent, easy to remove and clean. For young children, playful or colorful aesthetics may increase compliance [118]. Simple hook-and-loop or buckle closures make donning and doffing devices easy for caregivers. Some pediatric populations will have sensory sensitivities and, for them, strapping that is soft and compliant but secure needs to be available. Devices are ideally waterproof, which can prevent recording interruptions during hand washing or bathing, and data capture can continue during swimming or other water-based activities. If these special considerations to implementation are overcome, wearable device systems can bring uniquely powerful diagnostic and prognostic information into pediatric care, a setting where real-world, objective assessment is otherwise very difficult.

#### 3.5.2. Individuals with Cognitive Deficits

Great potential exists for the use of wearable devices in the routine clinical care of individuals with cognitive impairment. For example, ankle-mounted accelerometers have been used to distinguish between healthy controls and participants with Alzheimer’s disease prior to the onset of major clinical behavioral impairments [119]. Cognitively impaired individuals cannot always accurately report on their performance in daily life and proxy-reports from caregivers may lack accuracy [120]. Wearable devices can avoid these issues by providing clinicians with objective, real-world data.

Several special considerations arise from the individual’s cognitive impairment. Those individuals with more severe impairment may not understand the purpose of a wearable device. Devices that are small, light, and comfortable are more unobtrusive and will help avoid resistance to wear [3]. Device discomfort has led to as much as 25% withdrawal from studies of a neurological population [121]. The level of caregiver support available will be critical for success, as individuals with severe impairment may not be able to communicate discomfort (e.g., excess tightness, skin irritation) during device use. Device design could facilitate comfortable wearing by employing soft but strong materials and avoiding hard edges near areas of fragile skin. There will be a trade-off between the simplicity to don and doff by the patient themselves [3,121] versus minimizing the proclivity to remove the devices when it is not appropriate to do so. An option of a more secure clasp that requires caregivers or clinicians to remove may be appropriate. When possible, wearable devices will need to be attached to the patient and not their clothing, to avoid inadvertently laundering or discarding the device [121]. Wearable devices with long battery life could remove the need for cognitively-impaired individuals and/or caregivers to remember to monitor and charge the battery. Devices that are aesthetically pleasing may facilitate compliance [3], such as the emerging devices designed to appear as watches or jewelry (e.g., Motionlogger Micro Watch, AMI, Ardsley, NY, USA; Motiv Ring, Motiv, San Francisco, CA, USA). Lastly, cognitive impairment heightens the need for clear caregiver education related to the device protocol and interpretation of results, preferably with simple graphical instructions and data visualizations.

## 4. Benchmarks for Future Development

### 4.1. Proposed Benchmarks

While there are serious barriers to implementation of wearable device systems into routine physical rehabilitation practice, there is also substantial opportunity for growth and development in the nascent wearable sensor technology field. Here, we propose benchmarks for wearable device systems that might be readily adopted into clinical practice. Figure 3 is a visual illustration of the barriers and benchmarks. Each barrier is represented by a circle, with barriers that will be more difficult to overcome indicated by larger circles and hotter colors. Once barriers are overcome (circles shrink enough to fit through the funnel), then wide-spread clinical implementation (green rectangle) will be possible. The specific benchmarks (Table 2) are intended to serve as a guide for engineers, software developers, clinician-scientists, and clinicians alike as we pursue this important goal. They arise from the 10+ year history of using wearable device systems in our laboratory with more than 400 research participants, our clinical practice experience, and discussions with a range of rehabilitation clinicians.

As with any technology, wearable device systems that are commercially-available with comprehensive, accessible technology support for end-users (rehabilitation professionals and patients) are necessary for routine implementation into clinical practice [122]. Moving a system from laboratory development into commercial production therefore becomes an important goal for system developers. System developers and their manufacturing partners will need to ensure that any systems sold as medical- (or research-grade) devices will need to meet the clinical safety standards defined by the International Organization of Standards (ISO/TC 173 for Assistive Products and ISO 10667-1:2011 for Assessment Service Delivery) and evaluated by national or multi-national regulatory agencies. Ideally, a system would be marketable to both clinicians and direct to consumers, as that would facilitate uptake [59,123]. While it is possible that research-grade device systems that are compliant with clinical regulatory standards can be adopted into clinical care, systems that are sold directly to patients and other consumers will better enable the wide-spread, ubiquitous adoption that would be most desirable. Time benchmarks flow from the previously discussed need to maximize patient care time and minimize undue burden for busy clinicians (Figure 1 and Figure 2). While many, varying protocols for wearing duration may be used going forward, a device that can be comfortably worn for 12–24 h at time would be an excellent staring point and would facilitate adherence to wearing. A device system that works as intended/programmed for every patient is a prerequisite for wide-spread clinical adoption [59]. If devices are fickle and do not work (i.e. fail to record, store, or upload data) even a small percentage of the time, rehabilitation clinicians and patients will become frustrated and discontinue use. The most challenging and time-consuming benchmarks to achieve will be accurate algorithms and standardization of variables of interest that are reliable, valid, and responsive in the heterogeneous physical rehabilitation clinic populations. Since clinicians typically carry patient caseloads that include more than one patient population (e.g., Monday morning caseload includes a person with stroke, a person with multiple sclerosis, and a person with Parkinson disease), algorithms that work across populations and the ability to select specific variables of interest will make it easier to implement a wearable device system into daily practice. Finally, consumer-friendly reports that are quickly understandable to clinicians, patients, and families and integrated into the electronic medical record will allow the movement performance data collected from daily life to be used for shared clinical decision making and motivation.

### 4.2. Example Application of Benchmarks to A Currently-Available System

This section applies the benchmarks to a currently-available wearable device system as an example. The example system is the Actigraph wGT3X-BT device paired with the ActiLife software (Actigraph Inc, Pensacola, FL, USA). This system is one of the most commonly used in North America. Figure 4 is a picture of the wearable devices on a participant (Figure 4A) and a screen-shot of the software interface (Figure 4B). The wGT3X-BT contains a solid state, triaxial, microelectromechanical system (MEMS) accelerometer, an ambient light sensor, and a capacitive proximity sensor to detect wear time. It measures 4.6 × 3.3 × 1.5 cm and weighs 19 g.

Sampling rates can be set within a 30–100 Hz range, with slower sampling rates enabling longer data collection periods. The accelerometer has a dynamic range of +/− 8 g. Memory is 4 GB, with a battery life of 25 days and data storage possible for up to 180 days. The device has Bluetooth communication capability, which when activated, reduces battery life.

We benchmark this wearable device system’s use for capturing upper limb performance in daily life, indexed by the use ratio variable in Table 3. As briefly mentioned in the previous section, the use ratio is a variable quantifying the relative duration of movement in one limb to the other. Determination of benchmark achievement was decided based on ours and others experience with the wearable device system (convenience, time, ease, comfort, device operations, and reporting benchmarks) and the references included in the prior section (algorithm and standardization benchmarks).

As can be seen in the right column of Table 3, around half of the proposed benchmarks have been achieved to date for this wearable device system and variable of interest. We respectfully note that this system is targeted for researchers and not for rehabilitation clinicians and patients. The conclusion drawn from Table 3 is that this wearable device system and its corresponding variable of interest does not yet meet all the benchmarks proposed here for widespread implementation into routine clinical rehabilitation practice. This variable of interest first appeared in published literature in the year 2000. Similarly, the wearable device system has been commercially available for more than 10 years and is the device system that is most commonly used in rehabilitation research studies. Thus, even a system that is commonly used by researchers is not yet ready for clinicians and patients.

Many new commercially- and laboratory-available wearable device systems are introduced each year. We are unable to benchmark a newer device system for this review because new systems, by default, have limited publications about how they operate and the variables that are derived from them. Publications from commercial entities are often not available at all, or come in the form of white papers that are not peer-reviewed. Publications from research laboratories about brand new device systems often report information about the device system and testing in a few healthy, young participants and/or a few patients, but do not yet have sufficient information/detail for assessing progress towards achievement of the proposed benchmarks for adoption into clinical care. As can be seen in this benchmarking exercise, it takes a long time for new devices to be picked up by the clinical research community and the generation of subsequent publications documenting clinical feasibility and standardization of variables. Nonetheless, the benchmarking exercise serves as a useful example of the collective progress that needs occur.

## 5. Conclusions

In the foreseeable future, the promise of wearable sensor technology for improving physical rehabilitation practice will be realized (Figure 3). A key goal for the field is to work to move wearable device systems that measure human movement into routine clinical practice. Major barriers to implementation arise from both current clinical practice and from the wearable device systems themselves. Clinical practice barriers include the busy clinical environment and a not-yet full realization of the value of the critical information that can be obtained. Wearable device system barriers include: (1) consumer-grade devices that are not accurate for many physical rehabilitation patient populations; (2) research-grade devices that are not user-friendly for clinicians or patients; (3) insufficient published data regarding reliability, validity, and responsiveness of output variables that can inform clinical decisions; and (4) the need to have these data on a range of output variables so that clinicians can select the most appropriate ones for specific patients.

As noted in the Introduction, we are at the beginning of this effort. Next generation wearable sensors and device systems will be smaller, faster, and allow enormous flexibility for clinicians and patients to gather data via wireless sensors networks and secure cloud technology, facilitating widespread adoption. Comparing the current status of the field to the development of telephones through smart phones, we are entering the early 1900s, when telephones were present and available in select locations, and switchboard operators were required to make connections between parties. Similarly, wearable devices systems have become increasing common in the research-world and require research-trained personnel to make them work. The collective efforts of engineers, computer scientists, clinician-scientists, and clinicians can push us all towards a future with sleek, ubiquitous, and easy to use wearable device systems in physical rehabilitation practice.

## Figures and Tables

**Figure 1 sensors-20-05744-f001:**
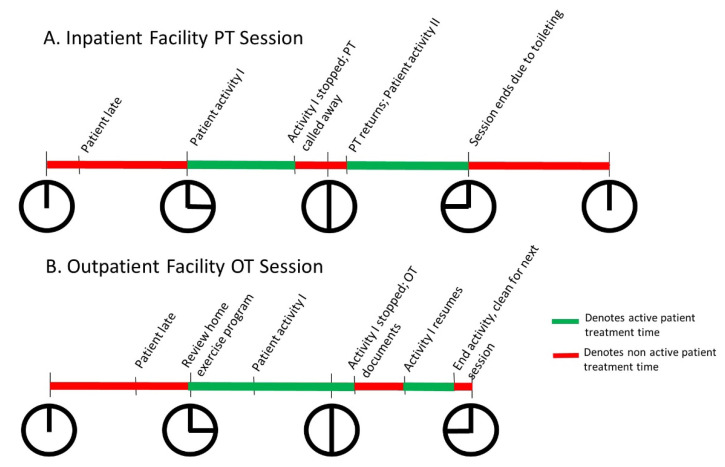
Clinicians encounter many barriers to spending time in direct patient care. Examples of inpatient physical therapy (PT) 1 h session (**A**) and outpatient occupational therapy (OT) 45 min session (**B**).

**Figure 2 sensors-20-05744-f002:**
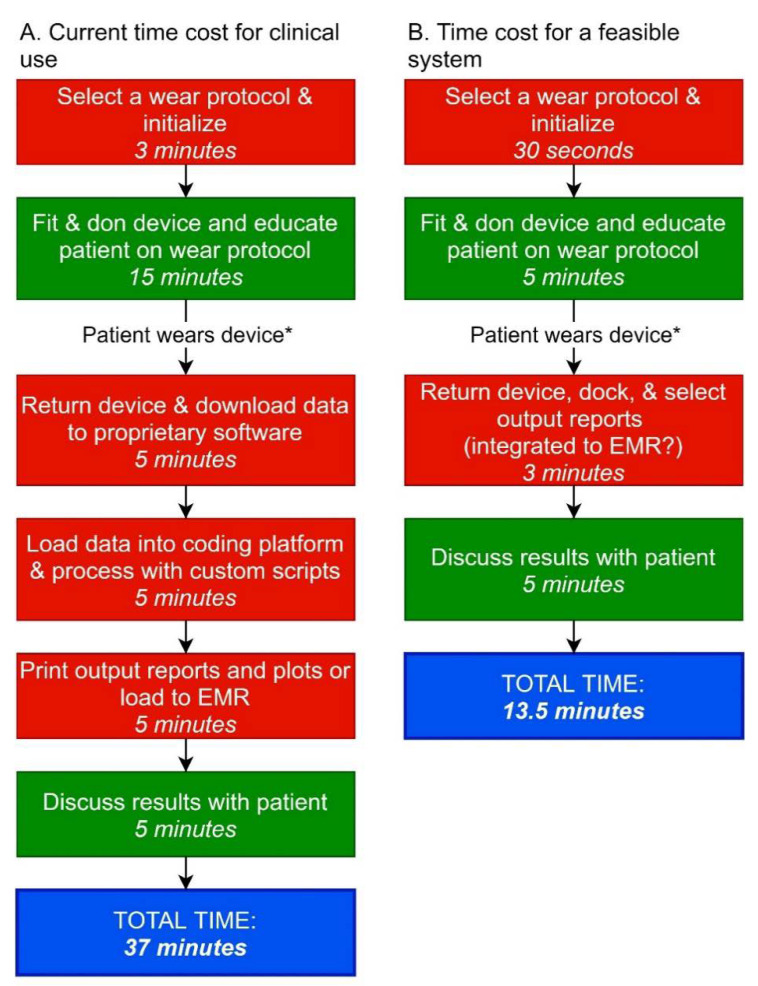
Time Cost for Clinical Implementation of Wearable Device Systems. **A**: Current time cost in clinical practice. **B**: Estimated time cost for realistic implementation into clinical practice. The first two boxes in **A** & **B** are estimates for an experienced clinician using a wearable device system with a new patient. Red boxes indicate non-billable time while green boxes indicate billable time. EMR: electronic medical record. * Wear time may vary from hours to several days.

**Figure 3 sensors-20-05744-f003:**
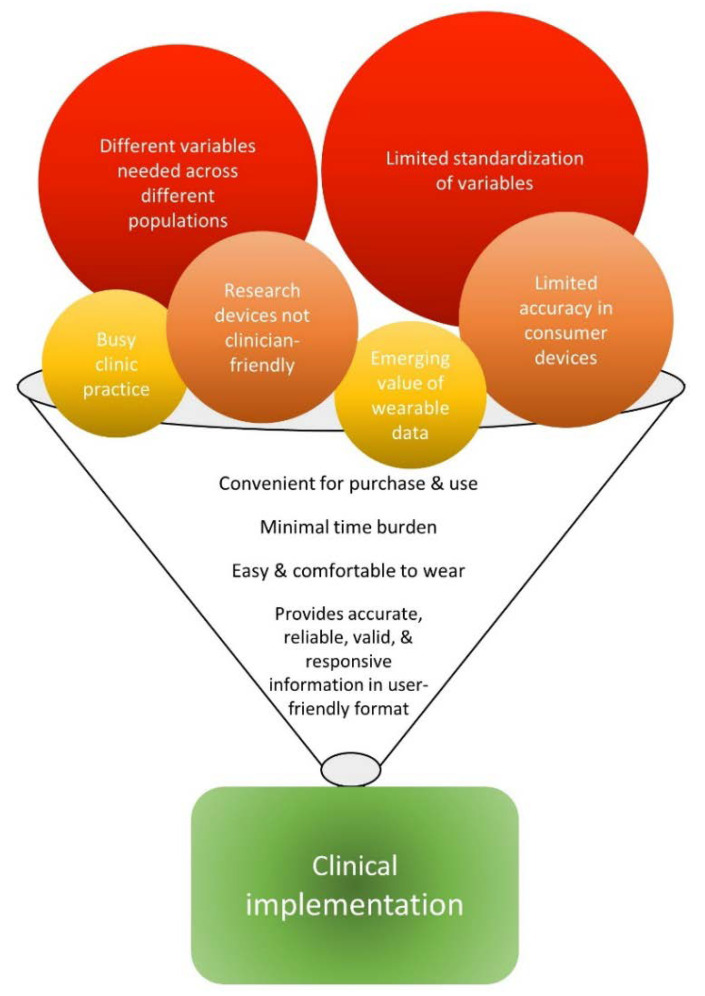
Barriers and benchmarks for implementation of wearable sensor technology into routine clinical rehabilitation practice. The warmest colors and largest circles indicate barriers that will take more work to overcome. Once the barriers are sufficiently reduced, then widespread clinical implementation (green rectangle) will be possible.

**Figure 4 sensors-20-05744-f004:**
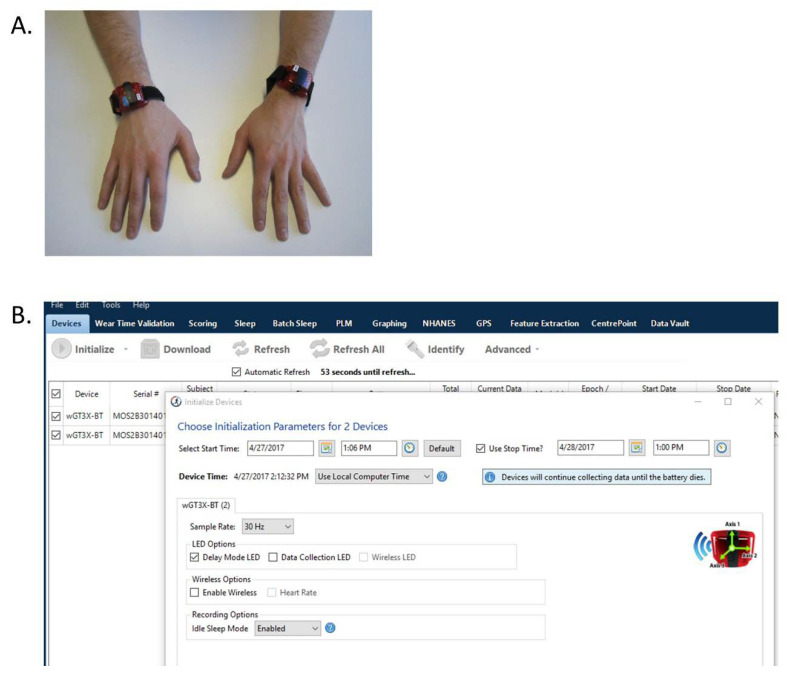
(**A**): Photo of a participant wearing the Actigraph wGT3X-BT devices on each wrist. (**B**): A screen shot of the ActiLife software interface used to interact with the wearable device. The software provides options to change various recording parameters (start and stop times, sampling rate, etc.) prior to recording/collecting data.

**Table 1 sensors-20-05744-t001:** A sample of wearable-derived variables proposed in the literature. This list is not exhaustive, with new variables proposed all the time. Published variable data have been judged as: Green = sufficient data for implementation into clinical practice; Yellow = some data available to-date, but not yet sufficient for implementation; and Red = no data yet.

Variable Name	Explored In:		Evaluation in Health Condition:		
	Absence of Health Condition	Health Condition	Reliability	Validity	Responsiveness
**Lower Limb** [16,18,21,30,34,40,41,49,53,57,64,65,66,67,68,69,70]					
*Time-based variables*					
% time inactive	●	●	●	●	●
Walking duration	●	●	●	●	●
*Amount-based variables*					
Steps/day	●	●	●	●	●
Bouts/day	●	●	●	●	●
Steps/bout	●	●	●	●	●
*Intensity-based variables*					
Stepping intensity	●	●	●	●	●
Maximum output	●	●	●	●	●
Mod. intensity minutes	●	●	●	●	●
Peak activity index	●	●	●	●	●
*Other variables*					
Step length variability	●	●	●	●	●
**Upper Limb** [21,32,71,72,73,74,75,76,77,78,79,80,81,82,83,84,85,86,87,88,89,90,91,92,93,94,95,96]					
*Time-based variables*					
Hours/duration of use	●	●	●	●	●
Use/activity ratio	●	●	●	●	●
*Amount-based variables*					
Acceleration area	●	●	●	●	●
Activity counts	●	●	●	●	●
Mono-arm use index	●	●	●	●	●
*Intensity-based variables*					
Acceleration variability	●	●	●	●	●
Acceleration magnitude	●	●	●	●	●
Acceleration asymmetry	●	●	●	●	●
Laterality index	●	●	●	●	●
Magnitude ratio	●	●	●	●	●
Bilateral magnitude	●	●	●	●	●
*Other variables*					
Variation ratio	●	●	●	●	●
Jerk asymmetry	●	●	●	●	●
Spectral arc length	●	●	●	●	●

**Table 2 sensors-20-05744-t002:** Benchmarks for wearable device systems that will facilitate implementation into routine physical rehabilitation clinical practice.

	Benchmark
Convenience for purchase and use	Commercially-available, consumer-grade device system that can be easily used by clinicians and consumers; comprehensive, accessible tech support.
Initial set-up time for clinician	5–6 min for first time with new patient.
Routine set-up time for clinician	≤1 min in subsequent times with same patient.
Time to extract data and generate output or report	≤5 min
Ease of donning/doffing for patient	≤2 min; without assistance from another person if intended for home use.
Comfort for extended wear	Soft plastic or other flexible strapping that can be tolerated 12–24 h/day; no hard edges on device that push into skin; water resistant so does not have to be removed for bathing, dishwashing, etc.
Device operations	≥95% of the time, device collects, stores, and/or uploads data as programmed and does not malfunction.
Algorithms for extracting data and generating variables of interest	≥90% accuracy to measure intended construct; must be accurate across a broad range of movement abilities typically seen in physical rehabilitation clinics.
Standardization of variables of interest	Reliability: consistently captures construct with reliability coefficients of ≥0.80.
	Validity: comprehensively captures construct that has known relevance to clinical decision-making and management.
	Responsiveness: detects changes of ≥5%; changes of 5–10% or higher provide relevant information for clinical decision-making and management.
	Values can be computed & reported in sensor-independent units.
Report to clinician and patient	Consumer friendly, targeting audience with ≤ secondary school education; 1–3 key outcome variables presented; simple graphics with colors to make accessible across languages and language and/or cognitive deficits; ability to integrate into electronic medical record.

**Table 3 sensors-20-05744-t003:** Application of benchmarks to a current wearable device system, the Actigraph wGT3X-BT and ActiLife software for measuring upper limb performance in daily life, indexed by the use ratio variable.

	Progress toward Benchmark
Convenience for purchase and use	*Commercial-availability achieved*. Can be easily purchased. *Consumer-grade not achieved*. Marketed and sold as research-grade device. Technology support helpful for researchers but would be too difficult for clinician or patient consumers.
Initial set-up time for clinician	*Not achieved*. Current time estimate is 18 min.
Routine set-up time for clinician	*Not achieved*. Current time estimate is 8–10 min.
Time to extract data and generate output or report	*Not achieved*. Current time estimate, using ActiLife + custom-written software in MATLAB or R is 15 min
Ease of donning/doffing for patient	*Achieved*. Can be done at home for most patients without assistance from another person.
Comfort for extended wear	*Achieved*. Allows for variety of strapping options and has been worn 12–24 h by hundreds of patients, with many wearing it for 24 hrs 1x/wk or 1x/month. Water resistant.
Device operations	*Achieved*. Have lost data <2% of the time.
Algorithms for extracting data and generating variables of interest	*Achieved for use ratio*. Algorithm is stable across a range of movement abilities in typical adults and children, and persons with stroke.
Standardization of variable of interest: Use ratio	*Reliability achieved*. Test-retest reliability coefficient = 0.86 [79]
	*Validity achieved for adult stroke population, but not other populations* [99]. Captures relative use of the upper limbs, which is stable and narrowly distributed in referent populations [83,88], but wide-ranging post stroke.
	*Responsive to change achieved*. Can detect changes of ≤5% [32,79]. *Clinical relevance of change not achieved*. Currently unknown how much change is clinically meaningful.
	*Sensor-independent units achieved*. Values are a ratio, making differences across sensors irrelevant.
Report to clinician and patient	*Not achieved*. Current output can be consumed by trained researchers but is not clinician-, patient-, or family friendly. Output is not integrated with electronic medical record.

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
