# Peer review of "Implementation of Wearable Sensing Technology for Movement: Pushing Forward into the Routine Physical Rehabilitation Care Field"

_sensors, 2020, doi:10.3390/s20205744_

Round 1

Reviewer 1 Report

The authors have brought an interesting topic of wearable sensing tech to be pushed into Routine physical rehabilitation. The following aspects regarding the present manuscript have to be addressed:

  1. Noticed the manuscript talks about motion sensor based wearable sensing technology. The title doesn't reflect the contents from the context.
  2. Since medical/clinical devices/systems are regulated products, an actual barrier is how to make these identified wearable devices to meet the clinical standards, i.e. ISO. The front-line clinical professionals for impatient facility and outpatient facility even the users at home, then will adopt these wearable devices.
  3. The purpose of any domestic grade devices is far way from the regulated product standards in term of accuracy, reliability and applicability. surprisingly, why the authors haven't accounted for this type of wearable devices into the rehabilitation environment.
  4. As long as the research grade wearable devices are along with or in compliance of clinical rehabilitation standards. the adoption of these will become possible.
  5. There are some minor issues of English grammar. please use present tense for the authors to state the facts and the outcomes. 
  6. Although the authors have well analysed the present situation of physical rehabilitation with affording little time (see Fig 2), the requirements / or specifications of rehabilitation for inpatient and outpatients are much more than this based upon the clinical practice standards. Hence the angel from affording time to push wearable techs forward for routine physical rehabilitation care isn't a comprehensive solution in particular for the review work.

Reviewer 2 Report

Overall, I find the paper is well-executed and to be useful for researchers and developers in wearable health monitoring systems. I recommend the paper be published in the Sensors journal.

There are a few items that need attention. Specific comments needing attention are provided below.

It would be useful to add information about:

-a new generation of wearable sensors and systems that allows clinicians to gather measures by means of the wireless sensor network and cloud technology;

-the advantages of performing physical parameter measurements using equipment connected to Internet by comparison with the conventional ones are: the storage of data in a remote database accessible by Internet; the data analyze of measurements; the perform of specific actions based on the results of these measurements;

-a tendency to make medical device IoT compatible, that may contribute to the quality and effectiveness of the healthcare services

- the recent progresses in physical rehabilitation care of the some commercial wearable device systems together with the corresponding structural scheme or foto. In Example application of benchmarks to a currently-available system part (chapter 4.2), a currently-available wearable device system of Actigraph Inc. should be explained together with the corresponding structural scheme and basic information of the device.

Reviewer 3 Report

The manuscript discusses the current limitation of wearable device in clinical practice and aims to make a standard for using data from wearable device.

In reviewer idea, the information presented in the paper is of interest to both clinical practitioner and device maker. However, there are some questions that need to be addressed before the manuscript can be recommended for further processing.

  1. As mentioned in abstract, the focus is physical rehabilitation, therefore it is needed to highlight what type of wearable devices which are discuss here. From what I can see in the manuscript, probably authors want to discuss about activity measuring device rather than other physiological devices, such as heart beat sensor, oxygen sensor.
  2. Also, what type of information or parameters to be measured should be mentioned so that reader will have general idea of what to expecting for a “wearable device” discussed in this paper.
  3. Section 2.2: In a general treatment scheme, which parameter will be monitored? How does it be measured in the current clinical practice?
  4. Section 4.4: In the previous section, author was discussing about standardization of the data from different device; however, this step is not included in the proposed benchmark scheme.
  5. Figure 3: the idea present in the figure is not easy to understand. I am confused of the reason of why did author choose circle shape represent for barriers, what is the cone shape for? Why does Clinical implementation” have that “star shape”?
  6. Table 3, in the evaluation, there are “achieved” and “not achieved” to represent for the progress. What is “partially achieved”? How to evaluate how much “partially” has been fulfilled?
  7. Table 3: a lot of evaluation has been given for the device. But reader might question that what is the final conclusion, and how to draw that conclusion from the table?
  8. Section 4.2, can authors try the benchmark with another device which is introduced recently? Because evaluation of a device from 2000s seems to be not enough for a tool during nowadays, when a lot of devices are introduced every year.

Round 2

Reviewer 1 Report

Noticed that the authors have mostly answered the questions arisen from the comments with a certain amount of literature reviewing work.

However, "Standardization of the data" hasn't be referred by the formal standards as indicated in the previous comments, i.e. relevant ISO etc. Can the authors add up?

The font resolutions of Fig1,2,3 and 4 are poor for readers to go through the phrases or sentences.

Reviewer 3 Report

The response from authors has mostly answered reviewer's concerns.  Reviewer understand that the field is new and should be explored more.
The manuscript can be recommended for publication.
